# Water Resources Carrying Capacity Based on the DPSIRM Framework: Empirical Evidence from Shiyan City, China

Wenming Cheng [1], Jing Zhu [1,*], Xiaochun Zeng [2], Yuan You [1], Xuetao Li [1] and Jun Wu [3]

1  School of Economics and Management, Hubei University of Automotive Technology, Shiyan 442002, China;
   chengwenming009@163.com (W.C.); yy18872132461@163.com (Y.Y.); lixt_jg@huat.edu.cn (X.L.)
2  School of Economics and Management, Xi'an University of Technology, Xi'an 710000, China;
   zengxc1107@163.com
3  School of Mathematics, Physics and Optical Engineering, Hubei University of Automotive Technology,
   Shiyan 442002, China; wjglo@huat.edu.cn
*  Correspondence: 20150015@huat.edu.cn

**Abstract:** In this article, we construct an evaluation index system based on the DPSIRM framework to determine the water resources carrying capacity of Shiyan City. Then, we use an obstacle model to calculate and analyze the factors that constrain the improvement in the water resources carrying capacity in the city. The research results are as follows: (1) The water resources carrying capacity of Shiyan City was on the rise during 2011–2021, and the water resources carrying capacity of Shiyan City was continuously improved. (2) The management system is the primary obstacle subsystem, followed by the driving force system, the response system, the pressure system, the state system, and the influence system. (3) Among the specific factors, the top three obstacles are sewage treatment investment, the proportion of guaranteed harvest area in drought and flood, and the average annual fertilizer applied per unit of cultivated land. These primary factors restrict Shiyan City from improving its water resources carrying capacity. This study has important practical significance for understanding the resilience of the water system in Shiyan City; exploring the changes in the water resources carrying capacity and its obstacle factors; and guiding the development, utilization, and management of water resources in Shiyan City.

**Keywords:** water resources carrying capacity; DPSIRM framework; Shiyan; resilience



## 1. Introduction

Water is vital in crop production, economic activity, and sustainable ecological development [1]. Water resources are essential in supporting sustainable economic and social development and maintaining ecological balance [2], and they are irreplaceable in environmental and food security [3]. With the growth of the population and economic development, the problem of water shortages is becoming more and more serious, which is restricting the sustainable development of the region [4]. Water resources have become one of the most important resources in the process of economic and social development in the world. The carrying capacity of water resources directly affects the economic and social development of a region. The strength of a city's water carrying capacity reflects the resilience of the local water system. The stronger the carrying capacity of water resources, the higher the resilience of the water system. Evaluating the water resources carrying capacity of a region or city is of great significance for understanding the current situation of water resources in a region or city, understanding the resilience of a region's water system, and has a profound impact on how a city or region formulates water policies.

In order to conduct in-depth and quantitative research on the resilience of the water system in Shiyan City, this paper constructs the water resources evaluation index system of Shiyan City based on the DPSIRM framework. Then, by searching the data of the Shiyan Statistical Yearbook and Shiyan Statistical Bulletin, the entropy weight method,

comprehensive evaluation model, and barrier model were used to study the water resources carrying capacity of the city.

## 2. Literature Review

### 2.1. The Concept of Carrying Capacity

With the increasing contradiction between social development and resource shortages, the concept of the 'resource carrying capacity' was put forward by UNESCO in the early 1980s [5]. The carrying capacity refers to the maximum number, density, or biomass of people that a given area can support continuously [6]. Carrying capacity refers to the marginal capacity of a habitat or environment to provide the resources needed to sustain human life [7].The concept of carrying capacity of relative resources was first proposed by Huang and Kuang, which refers to 'calculating the relative resource carrying capacity of a research area based on the per capita resource occupancy or per capita consumption of the reference area and the resource stock of the research area based on one or more specific regions' [8]. Scholars generally believe that the concept of water resources carrying capacity was first proposed by the soft science research group of water resources in Xinjiang, China, in 1989. Some studies consider the water resources carrying capacity to be the capacity to sustain a society with a good standard of living [9]. A stable stage of socio-economic development, a good ecosystem, and a stable quality of life are supported by the maximum carrying capacity of water resources for human activities [10].

### 2.2. The Factors Causing Water Resource Overload or Water Shortages

The reasons for water shortage or overload have attracted widespread attention. Many scholars have researched this aspect. For example, factors that can overload the water resources carrying capacity are the quantity and quality of water resources [11,12], climate change [2,13], population size [14], urbanization [2], industrialization [2], economic structure [15,16], and human activities [13]. In addition, war, political instability, poor management, inadequate national environmental policy planning, poor administrative capacity, high investment needs, and a lack of environmental awareness all contribute to water overload or scarcity [17]. The analysis of the causes of water shortage can help solve practical problems and related research topics.

### 2.3. The Evaluation Method of Water Resources Carrying Capacity

(1) Some scholars use the single-index method to study the carrying capacity of water resources. For example, the carrying index (CI) and index of water supply–demand balance (IWSD) were used to study Tianjin's water resources carrying capacity. In this study, relevant data from 2004 to 2008 were used for research, and the results showed that the efficiency of water resource utilization in Tianjin is relatively low. It is predicted that after implementing water resource protection policies in 2010 and 2020, the trend of water resource utilization in Tianjin will become more reasonable, and the WRCC of Tianjin will exceed the national average [10]. (2) Some scholars have used scenario analysis to study the water resources carrying capacity of Beijing. In this study, using data from 1986 to 2002 and assuming a comprehensive domestic water consumption of 75 $m^3$/(person·year) in Beijing based on the water resources used in Beijing around 2002, Beijing's urban population carrying capacity was defined as 5–6 million people. By 2003, the total population of Beijing had reached 14.56 million, with an urban population of 11.51 million. The actual population size had exceeded its carrying capacity by nearly double. The population exceeding its carrying capacity is the fundamental reason for the severe water shortage in the region [14]. (3) Some scholars have used multi-index or comprehensive index methods in their studies, such as the fuzzy comprehensive evaluation method used by Zhou, Z. et al. By establishing the evaluation index system of medium water quality safety and using the analytic hierarchy process (AHP) to weigh the evaluation indicators, a fuzzy comprehensive evaluation model was established. The model was applied to the Liantang Water demonstration base in Shenzhen, and the results showed that the method

has a small error, avoids subjective randomness, and conforms to the actual situation [18]. Lv, A. et al. used the fuzzy comprehensive evaluation method to evaluate the vulnerability of the WRCC system in China, and the results showed that the risk of WRCC occurrence in North China was higher than that in South China, and that in developed areas, it was higher than that in developing areas. Among them, the Beijing–Tianjin–Hebei region is at the highest risk [2]. Wang, G. et al. combined the fuzzy comprehensive evaluation with the system dynamics model to study the carrying capacity of water resources, the results showed that if the current development model were continued, the carrying capacity of water resources in Changchun would continue to decline and remain at a lower 'normal carrying' level. The carrying capacity of water resources in Changchun City can be improved by changing the production mode and proportion of the national economy. The rational allocation of water resources and strengthening water ecological protection can significantly improve the carrying capacity of water resources and keep it in a 'positive carrying' state while maintaining stable economic and social development [19]. Wang, Y. et al. selected various indicators related to water resources, society, and the economy and established a comprehensive evaluation index system with multiple indicators. Research has shown that due to rapid development and population expansion, there is a serious shortage and overload of water resources in Wuhan. The future development of Wuhan is worrying, and the same concerns apply to Ezhou. Other cities in the Wuhan urban agglomeration, such as Xiaogan, Huanggang, Qianjiang, and Tianmen, have greater potential for carrying water resources [20]. (4) Wang, Y.F. et al., taking the Shendong mining area as the research object, used the gray prediction model to predict the water demand of the economy–society–ecosystem in the mining area from 2020 to 2030 under different scenarios, and the results showed that the allocation structure of water resources in the mining area needed to be further optimized, and the scale of water use in the mining area could not adapt to its carrying capacity [21]. (5) Yang, J.F. et al. constructed a water resources carrying capacity evaluation model based on the system dynamics model and used this method to evaluate the water resources status of Tieling in different scenarios. The results show that given the constraints represented by water resources, GDP growth is expected to trend to the s-curve growth model; rapid population growth may lead to earlier and more severe water resource constraints [22]. Hu, G.Z. et al. built a system dynamics model based on the five systems of population, ecology, water resources, water environment, and water ecology and studied the North Canal Basin's water resources carrying capacity. It is estimated that the water environment and resource carrying rate will fall to 2.60 and 0.94, respectively, in 2025, while the water ecological carrying rate will remain stable at 10.98 [23]. Sun, Y. et al. took the five subsystems of the economy—population, supply and demand, land resources, water pollution, and management—as macroeconomic factors affecting the sustainable utilization of water resources and then used the system dynamics model to build a feedback loop and inventory flow chart of the system to simulate the changes in the water supply and demand situation and the future supply and demand gap from 2005 to 2020. The results showed that the water use efficiency in China would be significantly improved compared with that in 2005. By 2020, the gap between water supply and demand will reach 220 billion cubic meters, 4.8 times that of 2005 [24]. Feng, L.H. et al. simulated the water resources carrying capacity of Yiwu using the system dynamics method. If the current water supply level is maintained, the water supply of Yiwu will not be able to meet the requirements in the near future [9]. (6) Weng, X.R. et al. combined the economic, social, and ecological characteristics of using 26 specific evaluation indicators and evaluated and analyzed the carrying capacity of water resources in Chongqing via principal component analysis. The results show that the carrying capacity of water resources in Chongqing was continuously optimized and gradually enhanced from 2003 to 2017 [25]. Wu, F. et al. selected 13 indicators from the four aspects of the economy, society, environment, and water resources and analyzed the water resources carrying capacity of Huai'an City via principal component analysis. The results showed that the water resources carrying capacity of Huai'an City declined year by year from 2013 to 2019 [26]. Scenario simulation: Yang, Z. et al. designed five

scenarios, conducted a simulation analysis of the water resources carrying capacity of Xi'an, and determined the city's social, economic, water supply and demand, and wastewater discharge development from 2015 to 2020. If the current social development pattern is maintained, WRCC (0.32 in 2020) will change from 'normal' to 'poor' [27]. Yang, J.L. et al. calculated the water resources carrying capacity of the Three Gorges Reservoir Area from 2005 to 2020 using the variable fuzzy evaluation method. From 2005 to 2020, although the population and GDP of the urban agglomeration increased, the water supply capacity first increased and then decreased. From 2005 to 2020, the carrying capacity of water resources in the Three Gorges Reservoir area showed an increasing trend [28].

*2.4. Research Framework*

At present, the driving force–pressure–state–impact–response (DPSIR) framework is the most widely used in environmental assessment. The DPSIR framework was originally developed by the European Environment Agency in 1995 to provide decision makers with information on environmental indicators in response to the European Environment Agency's proposal on how to develop an integrated environmental assessment strategy. DPSIR is an extension of the PSR framework, adding driving force (D) and impact (I). The PSR framework was originally developed by the OECD to organize its work on environmental policy and reporting. These five aspects are logically causally related, which is a very good analytical framework for exploring the relationship between environment and socio-economic activities. This analytical framework was initially a qualitative evaluation system, and scholars later used this evaluation system quantitatively.

Later, some scholars applied the DPSIR framework to environmental assessment, ecological security, water resources management, sustainable development, air pollution, and other fields. The representative studies on water resources security, water resources management, and water system risk using the DPSIR framework are as follows. Using the DPSIR (Drive, Stress, State, Impact, Response) method, Borja, A. et al. studied the ecological quality and risk of local water bodies in estuaries and coastal waters in the Basque country (northern Spain) [29]. Skoulikidis, N.T. examined the state of the environment of 15 major Balkan rivers within the framework of DPSIR, arguing that the wars, political instability, and economic crises of the previous decades, combined with administrative and structural constraints, poor environmental planning and inspections, and a frequent lack of environmental awareness, have put great pressure on the rivers [17]. Sun, S. et al. established an evaluation index of the sustainability of water resources use based on the driving force–pressure–state–impact–response (DPSIR) model. The sustainable change in the water resources system in Bayannur City was evaluated comprehensively using these indexes. The results showed that the driving force of local water consumption increases due to social and economic development and the change in residents' consumption structure. During the study period, the pressure on the water system increased due to the increase in driving indicators, while the status of water resources continued to decline [30]. Vannevel, R. embedded the DPSIR framework in The Pentatope Model for Environmental Analyses and used the Governance by Actor–Subject Impact Assessment (GASI) as the interface. A tool for water treatment of PTM-GASI-DPSIR was proposed [31]. Given the research of these scholars, we affirm the contribution of the DPSIR framework in water resources assessment. These studies teach the logical thinking needed to analyze such problems and determine the change characteristics and advantages and disadvantages of the research object from the two parts of the subsystem and the whole. This is very helpful for water resources planning and management.

However, with the deepening of the research on this kind of issue, the water resources carrying capacity in a region is not only affected by the difference in natural resource endowment and the driving force of water consumption but also affected by local management measures. The impact of management on the water resources system is omni-directional, and management affects the driving force, pressure, state, and response. The DPSIR framework does not yet incorporate management systems in this analytical framework, and this

does not reflect well on the role of government authorities in water management. Including the management system in this framework, from the perspective of scientific research, will increase management indicators from the evaluation system, broaden the evaluation of water system security and the water resources carrying capacity, and create a more comprehensive evaluation and more objective and accurate evaluation results. Moreover, it is of great value to further understand the internal relationship of each system of water resources and propose solutions to the problems of water resources management from the management perspective. Adding management elements to the DPSIR framework becomes DPSIRM.

Existing research literature on the water resources carrying capacity has provided many contributions in terms of research ideas and methods, and there has also been a lot of research on the constraints of the water resources carrying capacity. However, the existing research on water resources carrying capacity and constraints on the water resources carrying capacity are often studied separately rather than in combination. Therefore, this study will combine the evaluation of the water resources carrying capacity with the restrictive factors of the water resources carrying capacity to uncover the main factors affecting the water resources carrying capacity from all possible aspects. With respect to the research area, Shiyan City is the core water source area of the South-to-North Water Diversion Project. The carrying capacity of water resources in the core water source area is related to the safety of the local water system and the project's sustainability. A search of literature from the Web of Science shows that research on the water resources carrying capacity of the entire Han River basin where Shiyan City is located, from the perspective of water resources management, is scarce. In this study, DPSIRM is used to objectively uncover the state of the local water resources and help local water resources management. With respect to the research content, the specific index selection of this study draws on many academic research results and combines the actual situation of local indicators, which can provide a reference for the index selection of the same type of research.

## 3. Materials and Methods

### 3.1. Study Area

Shiyan City spans a northern latitude of 31°30′ to 33°16′ and eastern longitude of 109°29′ to 111°16′, about 200 km from east to west and about 195.5 km from north to south. Shiyan is located in the middle and upper reaches of the Han River in Central China and northwest of Hubei Province, the Hanshui Valley in the Qinling–Ba Mountains, the Qinling Mountains in the north, and the Bashan River in the south. Located in Hubei Province, China, Shiyan City is the only regional central city in the adjacent areas of Hubei, Henan, Shaanxi, and Chongqing. Danjiangkou Reservoir is the starting point of the middle route of China's South-to-North Water Diversion project, and the source of water for the project is located in the middle of Danjiangkou City in Shiyan, Hubei province. The construction of this important water conservancy project will have an influence on the distribution of water resources. Shiyan City is the main water source, so it is necessary to evaluate the water resources carrying capacity of Shiyan City to understand the water resources situation of Shiyan City objectively. The study area is shown in Figure 1.

### 3.2. Dataset and Source

This study involves the water resources data, ecological environment data, economy and industry data, and other data of Shiyan City derived from the 'Shiyan Statistical Yearbook' from 2012 to 2022, Shiyan City Statistical Bulletin. The basis of indicator selection, determination of positive and negative signs, and acquisition and calculation of indicator data are shown in Table 1.

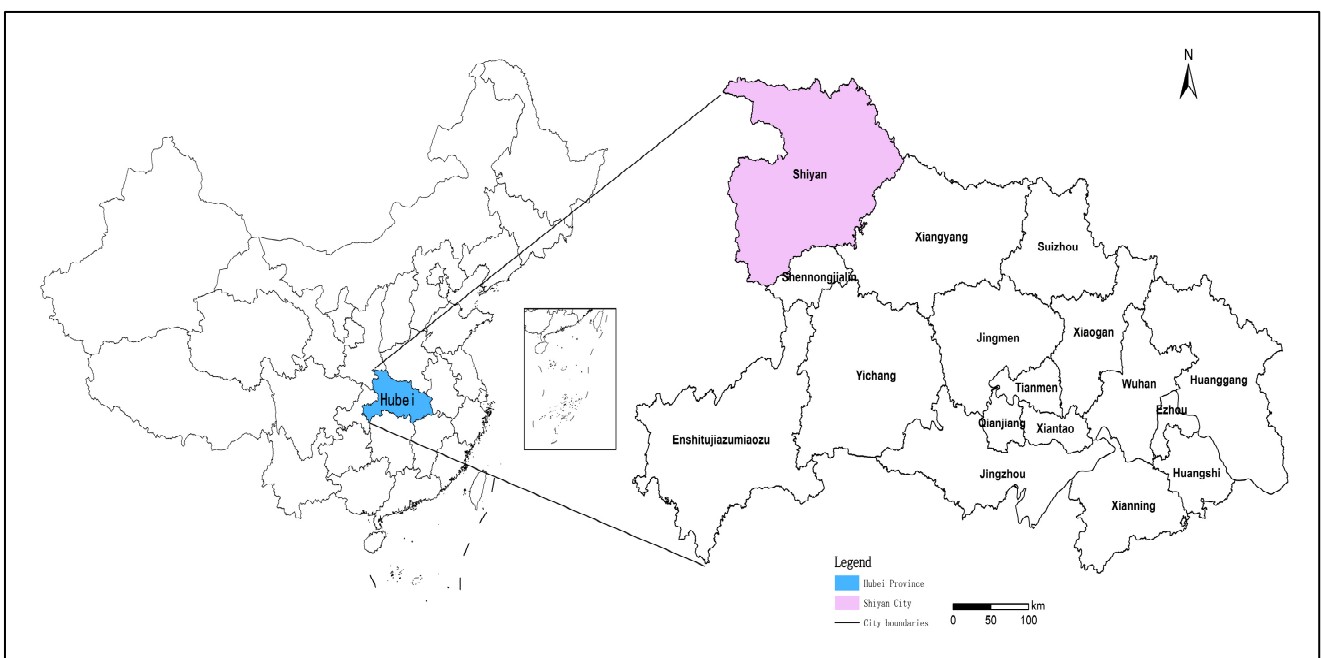

**Figure 1.** The study area covers parts of Hubei Province in China. Colored areas show the studied city.

*3.3. Methods*

3.3.1. DPSIRM Framework

The DPSIRM framework is the leading indicator system construction framework of current environmental assessment, which many scholars have recognized. The establishment of the DPSIRM framework has gone through a long development process, and the DPSIR framework is one of the original tools developed by the Organization for Economic Co-operation and Development (Paris, France) and the European Environment Agency for adaptive management of sustainable social environments (Copenhagen, Denmark) [32]. With the deepening of research in this field, some scholars have introduced the management subsystem based on DPSIR [33], from which the DPSIRM framework was gradually established and widely used. Based on Chai, N. and W. Zhou's research [34], combined with this study, the DPSIRM framework can be explained as follows.

The driving force subsystem (D) is the fundamental cause and initial driving force of changes in the regional water resources carrying capacity, mainly influenced by urbanization, economic and demographic changes, and development. The greater the driving force, the greater the risk of insecurity carried by water resources, and the greater the pressure and challenges faced by the resilience of water systems.

The pressure subsystem (P) refers to the pressure exerted on the water resources environment by intentional or unintentional human activities, such as the influence of the economic development level on water resources utilization, the change in water consumption of urban residents caused by urbanization development, and the discharge of industrial wastewater.

The state subsystem (S) refers to the actual changes and states of the regional water environment under pressure factors, mainly manifested as changes in the supply status, demand status, and water environment quality of water resources.

The impact subsystem (I) impacts the ecological environment, socio-economic, and other aspects, and the greater the impact, the less secure the regional water environment system.

Response subsystem (R) refers to the remedial actions that humans take to mitigate the negative impacts of the regional water environment, including responses to the treat-

ment of various pollution sources. The timelier the response, the safer the regional water environment system and the higher the resilience of the water system.

The management subsystem (M) represents people taking corresponding measures to control water resource management, such as formulating relevant policies and regulations and increasing financial investment in pollution control. Management is a proactive measure to enhance the carrying capacity of water resources and an important way to improve the resilience of water systems.

Based on the DPSIRM framework and considering the availability of data, an evaluation system of water resources carrying capacity in Shiyan City was constructed in this study, as shown in Table 1.

**Table 1.** Evaluation system of water resources carrying capacity in Shiyan City.

| Criterion Layer | Indicator Layer | Properties | Calculation Methods | Reference |
|---|---|---|---|---|
| Driving force (D) | $X_{D1}$ Per capita GDP (yuan) | Positive | From statistical data | [34,35] |
| | $X_{D2}$ density of population | Negative | From statistical data | [34,35] |
| | $X_{D3}$ urbanization rate | Negative | From statistical data | [34,35] |
| Pressure (P) | $X_{P1}$ Wastewater discharge per unit of industrial output value ($t$/10,000 CNY) | Negative | Amount of industrial wastewater discharge/industrial output value | [35,36] |
| | $X_{P2}$ Household water consumption (10,000 m$^3$) | Negative | From statistical data | [36] |
| | $X_{P3}$ Average annual fertilizer application per unit cultivated land (kg/hm$^2$) | Negative | Amount of fertilizer application/cultivated area | [35,36] |
| Status (S) | $X_{S1}$ Water resources per capita (m$^3$) | Positive | Amount of regional water resource/regional population | [34,35] |
| | $X_{S2}$ Water resources per unit area (m$^3$/hm$^2$) | Positive | Amount of regional water resources/regional land area | [34,35] |
| | $X_{S3}$ Annual precipitation (100 million cubic meters) | Positive | From statistical data | [37] |
| Impact (I) | $X_{I1}$ Proportion of guaranteed harvest area of drought and flood in cultivated land (%) | Positive | Guaranteed harvest area in drought and flood/cultivated area | [38] |
| | $X_{I2}$ Water quality in line with Class I~III standard proportion | Positive | From statistical data | [35] |
| | $X_{I3}$ Forest coverage rate (%) | Positive | From statistical data | [38] |
| Response (R) | $X_{R1}$ Sewage treatment rate (%) | Positive | From statistical data | [35,36] |
| | $X_{R2}$ Length of drainage pipe (km) | Positive | From statistical data | [34] |
| Management (M) | $X_{M1}$ Green coverage rate of built-up areas (%) | Positive | The annual built-up green cover area/green cover area | [34] |
| | $X_{M2}$ Investment in wastewater treatment (10,000 CNY) | Positive | From statistical data | [36] |

### 3.3.2. Index Weight Determination

The entropy method was used to determine the weights in this study. The main steps are as follows.

(1) Data standardization.

In order to eliminate the dimensionality of each index, data are standardized in this study. The formula for positive index processing is as follows:

$$MMS\_x_{ij} = (x_{ij} - Min) / (Max - Min)$$

The formula for negative index processing is as follows:

$$NMMS\_x_{ij} = (Max - x_{ij}) / (Max - Min)$$

(2)  Calculate the entropy of the j-th index.

$$e_j = k \sum_{i=1}^{n} x_{ij} \ln(x_{ij})$$

Xij here is the normalized data.

(3)  Calculate information on entropy redundancy.

$$d_j = 1 - e_j$$

(4)  Calculate the weights of each indicator.

$$w_j = \frac{d_j}{\sum_{j=1}^{m} d_j}$$

The calculation results of the water resources carrying capacity index weight are shown in Table 2.

**Table 2.** Index weight determination.

| Indicator | Information Entropy e | Information Utility Value d | Weight Coefficient w |
|---|---|---|---|
| MMS_D1 | 0.8854 | 0.1146 | 5.90% |
| NMMS_D2 | 0.7822 | 0.2178 | 11.20% |
| NMMS_D3 | 0.8522 | 0.1478 | 7.60% |
| NMMS_P1 | 0.9059 | 0.0941 | 4.84% |
| NMMS_P2 | 0.8889 | 0.1111 | 5.71% |
| NMMS_P3 | 0.8307 | 0.1693 | 8.71% |
| MMS_S1 | 0.9437 | 0.0563 | 2.89% |
| MMS_S2 | 0.9456 | 0.0544 | 2.80% |
| MMS_S3 | 0.9064 | 0.0936 | 4.81% |
| MMS_I1 | 0.8663 | 0.1337 | 6.87% |
| MMS_I2 | 0.9387 | 0.0613 | 3.15% |
| MMS_I3 | 0.9477 | 0.0523 | 2.69% |
| MMS_R1 | 0.9163 | 0.0837 | 4.31% |
| MMS_R2 | 0.9007 | 0.0993 | 5.11% |
| MMS_M1 | 0.8322 | 0.1678 | 8.63% |
| MMS_M2 | 0.7128 | 0.2872 | 14.77% |

Notes: MMS represents a positive indicator; NMMS represents a negative indicator.

### 3.3.3. Obstacle Degree Model

Step 1: Calculate F value. F value = W * P (W is the weight of the criterion layer, P is the weight of the indicator layer), and SPSSAU will calculate the weight of the indicator layer corresponding to the criterion layer after normalization processing by default.

Step 2: Calculate the standardized value of R′. The formula for calculating the standardized value of R′ is: (X − min)/(max − min); that is, (a datum − the minimum value of this index)/(the maximum value of this index − the minimum value of this index).

Step 3: Calculate I value; I value = 1 − R′ standardized value.

Step 4: Calculate the O value of the index layer. The formula is as follows:

$$O_j = \frac{F \times I}{\sum_{j=1}^{m} (F \times I)}$$

In this formula, j represents the indicator number; there are m-many indicators in total.

$$U = \sum O_j$$

Here, j indicates the subsystem number.

## 4. Results

### 4.1. Change in Comprehensive Score of Water Resources Carrying Capacity in Shiyan City

After calculating the weight according to the entropy value method, the comprehensive score change chart of water resources carrying capacity in Shiyan City was calculated, as shown in Figure 2. As can be seen in the figure, the water resources carrying capacity of Shiyan City showed an upward trend from 2011 to 2021 as a whole, indicating that the water resources carrying capacity of Shiyan City was continuously enhanced from 2011 to 2021. The resilience of the Shiyan water system also increased. There was a downward trend from 2011 to 2012 and from 2017 to 2018 and an upward trend in other years. During this period, the year with the weakest water resources carrying capacity was 2018, with a comprehensive score of 0.3870. The highest water carrying capacity was in 2021, with a composite score of 0.6547.

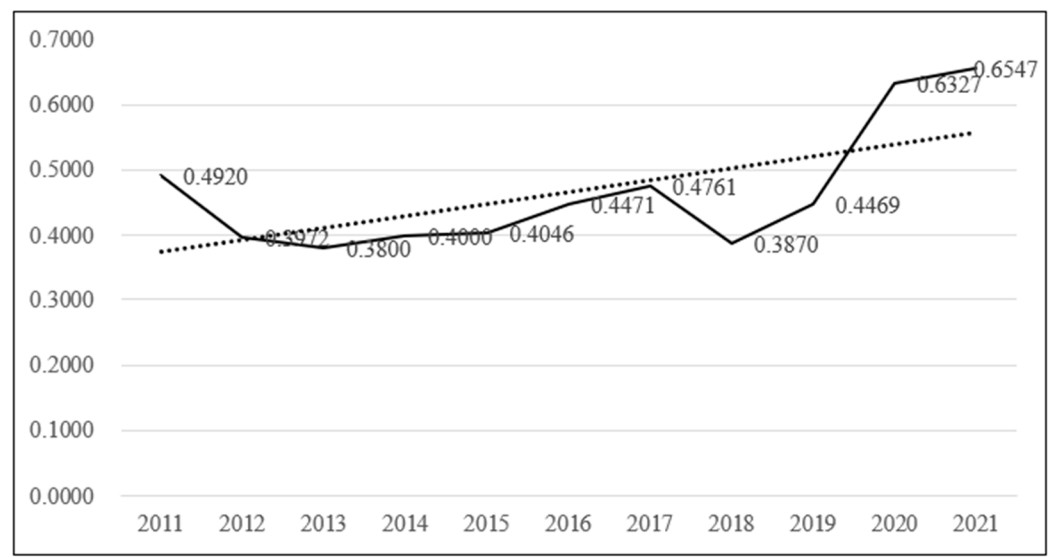

**Figure 2.** Change of water resources carrying capacity in Shiyan City. Note: The solid line is Comprehensive Score of Water Resources Carrying Capacity in Shiyan City; The dashed line is the trend line.

### 4.2. Analysis of Water Resources Carrying Capacity Obstacle Degree

#### 4.2.1. Subsystem Obstacle Degree Analysis

The obstacle degree model formula was used to calculate the obstacle degree of each subsystem of the water resources system in Shiyan City from 2011 to 2021 (see Figure 3 and Table 3). As can be seen in Figure 2, there are obvious differences in the variation trend of the obstacle degree of each subsystem from 2011 to 2021. The subsystems of the driving force, state, and response showed a fluctuating downward trend, while the subsystems of pressure, influence, and management showed a fluctuating upward trend. Driving the subsystem down can promote the resilience of water resources in Shiyan City. The decline in the state subsystem indicates that the ability of the water resources system to support regional economic and social development in Shiyan City is weakening, which implies that the resilience of the water system has a declining trend. The decline in the response subsystem will hinder the improvement in the resilience of the water resources system in Shiyan City. The rise of the pressure subsystem and influence subsystem indicates that the pressure of the resilience improvement in water resources in Shiyan City is increasing. The rise of the management subsystem can improve the resilience of the Shiyan water system.

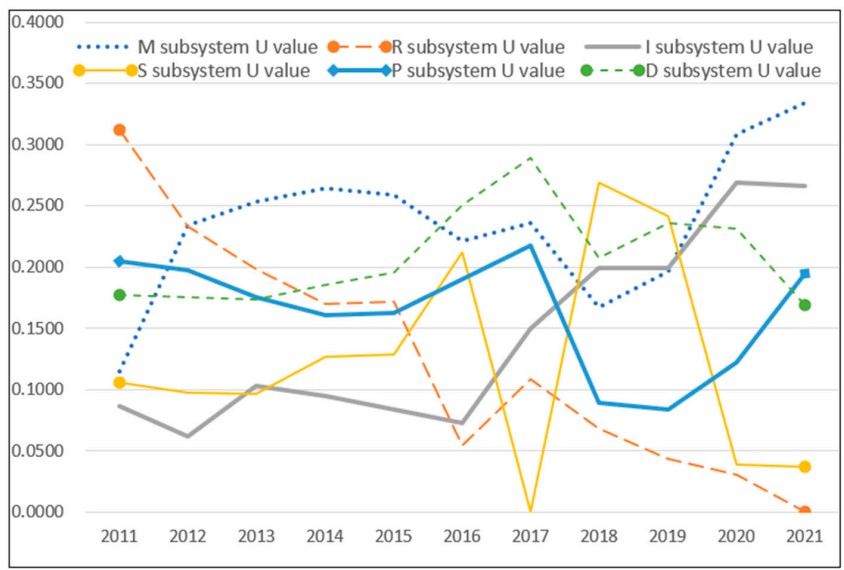

**Figure 3.** Trend chart of subsystem obstacle degree.

**Table 3.** Obstacle degree U value of each subsystem.

| Year | M Subsystem U Value | R Subsystem U Value | II Subsystem U Value | S Subsystem U Value | P Subsystem U Value | D Subsystem U Value |
|---|---|---|---|---|---|---|
| 2011 | 0.1149 | 0.3117 (I) | 0.0861 | 0.1056 | 0.2043 (II) | 0.1775 (III) |
| 2012 | 0.2343 (I) | 0.2329 (II) | 0.0618 | 0.0978 | 0.1975 (III) | 0.1757 |
| 2013 | 0.2533 (I) | 0.1987 (II) | 0.1028 | 0.0964 | 0.1757 | 0.1731 |
| 2014 | 0.2639 (I) | 0.1695 (III) | 0.0944 | 0.1264 | 0.1606 | 0.1852 (II) |
| 2015 | 0.2592 (I) | 0.1713 (III) | 0.0834 | 0.1283 | 0.1621 | 0.1957 (II) |
| 2016 | 0.2212 (II) | 0.0542 | 0.0727 | 0.2117 (III) | 0.1900 | 0.2502 (I) |
| 2017 | 0.2361 (II) | 0.1080 | 0.1492 | 0.0000 | 0.2173 (III) | 0.2894 (I) |
| 2018 | 0.1672 | 0.0679 | 0.1993 (III) | 0.2690 (I) | 0.0890 | 0.2077 (II) |
| 2019 | 0.1963 (III) | 0.0429 | 0.1997 | 0.2418 (I) | 0.0838 | 0.2356 (II) |
| 2020 | 0.3081 (I) | 0.0301 | 0.2687 (II) | 0.0390 | 0.1224 | 0.2317 (III) |
| 2021 | 0.3342 (I) | 0.0000 | 0.2658 (II) | 0.0370 | 0.1942 (III) | 0.1689 |

Notes: The parentheses indicate the ranking of obstacle subsystems, while (I), (II), and (III) represent the first, second, and third obstacle subsystems, respectively.

It can be seen in Table 3, under the DPSIRM framework, that the frequency of the occurrence of the top three subsystems of the obstacle degree of each subsystem under the overall framework from 2011 to 2021 is as follows: M system (nine times), D system (eight times), R system (five times), P system (four times), S system (three times), and I system (three times). The M subsystem was the first obstacle subsystem in 2012–2015, 2020, and 2021; the second obstacle subsystem in 2016 and 2017; and the third obstacle subsystem in 2019. This indicates that the management subsystem is the main constraining factor in improving the carrying capacity of water resources and the resilience of the water system in Shiyan City. In the future, further investment in water resource management is needed. The D subsystem was the first obstacle system in 2016 and 2017; the second obstacle system in 2014, 2015, 2018, and 2019; and the third obstacle system in 2011. This indicates that the population size, economic development level, and urbanization rate of Shiyan City from 2011 to 2021 are the factors that cannot be ignored when aiming to improve the water resources carrying capacity and enhance the water system elasticity of Shiyan City. The R subsystem was part of the top three obstacle subsystems from 2011 to 2015, with 2011 being the first obstacle subsystem, 2012 and 2013 being the second obstacle subsystem, and 2014 and 2015 being the third obstacle subsystem. After 2016, it was no longer part of the top three obstacle subsystems. This indicates that the response subsystem has undergone significant improvements in recent years, and it is recommended that Shiyan

City strengthen water resource management in its future development process. According to the characteristics of the local industrial structure of Shiyan City, it is important to adjust the water use structure, strengthen water-saving management, and improve the utilization efficiency of water resources.

4.2.2. Obstacle Degree Analysis of Each Factor

According to the obstacle degree model, the obstacle degree of the 16 influencing factors on the water resources carrying capacity of Shiyan City was calculated, and the top 3 obstacle degree factors were selected for statistical analysis. The results are shown in Table 4. The finding of the first obstacle showed that between 2011 and 2021, the M2 index appeared eight times, the R2 index appeared two times, and the D2 index appeared once. These results indicate that insufficient investment in sewage treatment was the main factor restricting the improvement in the water resources carrying capacity in Shiyan City from 2011 to 2021. According to the statistics of the second obstacle degree factor from 2011 to 2021, the I1 index appeared four times, the P3 index appeared two times, the R1 index appeared two times, the M2 index appeared once, the R2 index appeared once, and D2 index appeared once. This indicates that the I1 index is the second factor restricting the improvement in the water resources carrying capacity in Shiyan City, specifically referring to the proportion of guaranteed arable land in drought and flood. The guaranteed harvest area of droughts and floods refers to the effective irrigated area that can be irrigated in the case of a drought and discharged in the case of a flood. The higher the proportion of the guaranteed harvest area of droughts and floods, the lesser the impact of a change in the water resources carrying capacity on agricultural production. With regard to the second obstacle to improving the water resources carrying capacity in Shiyan, the proportion of the guaranteed area of droughts and floods is the most important restriction. According to the third obstacle, from 2011 to 2021, the P3 index appeared five times, the D2 index appeared once, the P2 index appeared twice, the S3 index appeared twice, and the M2 index appeared once. Regarding the third obstacle to improving the water resources carrying capacity in Shiyan, the average annual fertilizer application per unit of cultivated land is the most important. This shows that the agricultural fertilizer applied in Shiyan City has restricted the improvement in the water resources carrying capacity.

**Table 4.** Ranking of obstacle factors.

| Year | Category | No. 1 Obstacle | No. 2 Obstacle | No. 3 Obstacle |
|------|----------|----------------|----------------|----------------|
| 2011 | obstacle factors | R2 | R1 | P3 |
|      | obstacle degree | 0.1691 | 0.1426 | 0.1169 |
| 2012 | obstacle factors | R2 | M2 | P3 |
|      | obstacle degree | 0.1437 | 0.1403 | 0.1301 |
| 2013 | obstacle factors | M2 | R2 | P3 |
|      | obstacle degree | 0.1631 | 0.1339 | 0.1212 |
| 2014 | obstacle factors | M2 | P3 | D2 |
|      | obstacle degree | 01715 | 0.1127 | 0.1101 |
| 2015 | obstacle factors | M2 | R1 | P3 |
|      | obstacle degree | 0.1640 | 0.1237 | 0.1133 |
| 2016 | obstacle factors | D2 | P3 | M2 |
|      | obstacle degree | 0.1452 | 0.1233 | 0.1142 |
| 2017 | obstacle factors | M2 | D2 | P3 |
|      | obstacle degree | 0.2325 | 0.1679 | 0.1230 |
| 2018 | obstacle factors | M2 | I1 | S3 |
|      | obstacle degree | 0.1638 | 0.1454 | 0.1232 |
| 2019 | obstacle factors | M2 | I1 | S3 |
|      | obstacle degree | 0.1963 | 0.1623 | 0.1201 |
| 2020 | obstacle factors | M2 | I1 | P2 |
|      | obstacle degree | 0.3063 | 0.2490 | 0.1224 |
| 2021 | obstacle factors | M2 | I1 | P2 |
|      | obstacle degree | 0.3240 | 0.2658 | 0.1627 |

Further increasing investment in water resources and sewage treatment of Shiyan City is the primary measure to further improve the water resources carrying capacity of Shiyan City. The guaranteed harvest area of droughts and floods reflects the strength of the water resources irrigation system and drainage system in a region, and the improvement in the guaranteed harvest area of droughts and floods reflects the improvement in the water resources management system capacity in a region. The guaranteed harvest area is the area of farmland that can still produce a high and stable yield in case of a drought or flood disaster. It is a necessary measure for environmental protection and water resource protection to reduce the annual average fertilizer applied per unit of cultivated land.

## 5. Conclusions

The DPSIRM framework was used to study water system security from the perspective of water resources carrying capacity. From 2011 to 2021, the carrying capacity of water resources in Shiyan City gradually improved, as well as the safety of the water system. The resilience of the water system in Shiyan City also increased. The management system is the main obstacle subsystem, followed by the driving force system, the response system, the pressure system, the state system, and the influence system. Among the specific factors, the top three obstacle factors are sewage treatment investment, the proportion of guaranteed harvest area in drought and flood, and the average annual fertilizer applied per unit of cultivated land. These are the main factors restricting Shiyan City's water resources carrying capacity improvement.

In conclusion, although our study objectively evaluated the trend of changes in the water resources carrying capacity in Shiyan City, we also identified factors that constrained the further improvement in the resilience of the water resource system in Shiyan City from the perspectives of subsystems and specific factors. However, we did not fully consider the cross-regional spillover effects on water resource management in surrounding cities. Future research should expand the research area and further study the interactions and impacts between cities. In the selection of indicators in management, because there are no corresponding statistical data and no public multi-year data, only two dimensions were selected during the selection of indicators. This needs to be further enriched and perfected in future research.

**Author Contributions:** Writing—original draft preparation, W.C. and X.Z.; writing—review and editing, J.Z. and X.L.; data curation, Y.Y. and J.W. All authors have read and agreed to the published version of the manuscript.

**Funding:** This research was funded by the Doctoral Research Foundation of Hubei University of Automotive Technology (Grant No. BK202011), the Philosophy and Social Science Research Project of Education Department of Hubei Province (Grant No. 21Q167), the Hubei Key Research Base of Humanities and Social Sciences Open Fund (Grant No. WZ2021Y02), the Natural Science Foundation of Hubei Province (Grant No. 2022CFB959), the Educational Commission of Hubei Province of China (Grant No. Q20221802), the Hubei Key Laboratory of Applied Mathematics (Grant No. HBAM202105), and the Philosophy and Social Science Research Project of Education Department of Hubei Province (22Y112).

**Institutional Review Board Statement:** Not applicable.

**Informed Consent Statement:** Not applicable.

**Data Availability Statement:** Not applicable.

**Conflicts of Interest:** The authors declare no conflict of interest.

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
