# Peer review of "Water Resources Carrying Capacity Based on the DPSIRM Framework: Empirical Evidence from Shiyan City, China"

_water, doi:10.3390/w15173060_

Round 1
Reviewer 1 Report
Introduction
Line no-73, 75: Please add these words.
Line no-78: Please highlight the major achievements of all methods you mentioned, and describe the importance of DPSIRM model.
Materials and methods
Line no-134: This may be a pressure sub-system. Please make it more clear of the pressure on what?
Results
Line no-231: Please edit the year.
Line no-232: Please edit the year writing style.
Line no- 250: This is not an appropriate way of writing. Authors are requested to edit it like "The finding of first the obstacle ....".
Line no-267: This may be one time.
Line no-267: How did the authors calculate the average annual fertilizer application? Please mention in the method section and its source.
Conclusion
Line no-285: Please insert this whole paragraph in the introduction section.
Line no-290: Please write a short concluding result.
Line no-290: No need to mention the limitations in the conclusion section.

Dear authors,
The methods and result section need to be extensively reviewed. Do you agree to highlight a few result comparisons in the discussion section? Please make your concrete conclusion. It is very vague.
Author Response
Introduction
Line no-73, 75: Please add these words.
Response:Thank you for your valuable suggestion. The revisions are as follows:
In this study, relevant data from 2004 to 2008 were used for research, and the results showed that the efficiency of water resource utilization in Tianjin is relatively low. It is predicted that after implementing water resource protection policies in 2010 and 2020, the dynamic trend of water resource utilization in Tianjin will become more reasonable, and the WRCC of Tianjin will exceed the national average level.
Line no-78: Please highlight the major achievements of all methods you mentioned, and describe the importance of DPSIRM model.
Response:Thank you for your valuable suggestion. The main results and conclusions of all methodological studies are supplemented in this paper.The relevant literature of DPSIR research framework is added to the literature review, which complements the importance of DPSIRM framework.
At present, the Driving Force-Pressure-State-Impact-Response (DPSIR) framework is the most widely used in environmental assessment. The DPSIR framework was originally developed by the European Environment Agency (EEA) in 1995 to provide decision-makers with information on environmental indicators, in response to the European Environment Agency's (EEA) proposal on how to proceed with the development of an integrated environmental assessment strategy. DPSIR is an extension of the PSR framework, adding driving force (D) and impact (I). The PSR framework was originally developed by the OECD to organize its work on environmental policy and reporting. These five aspects are logically causally related, which is a very good analytical framework for exploring the relationship between environment and socio-economic activities. This analytical framework was initially a qualitative evaluation system, and later scholars used this evaluation system quantitatively.
Later, some scholars applied DPSIR framework to environmental assessment, ecological security, water resources management, sustainable development, air pollution and other fields. The representative studies on water resources security, water resources management and water system risk using DPSIR framework are as follows. Using the DPSIR(Drive, Stress, State, Impact, Response) method, Borja, A. et al studied the ecological quality and risk of local water bodies in estuaries and coastal waters in the Basque country (northern Spain)[32]. Skoulikidis, N. T. examined the state of the environment of 15 major Balkan rivers within the framework of DPSIR, arguing that the wars, political instability, economic crises of the past decades, combined with administrative and structural constraints, poor environmental planning and inspections, and often a lack of environmental awareness, have put great pressure on the rivers[33]. Sun, S. et al. established an evaluation index of sustainability of water resources use based on the drives-pressure-state-impact-response (DPSIR) model. The sustainable change of water resources system in Bayannur City was evaluated comprehensively by using these indexes. The results show that the driving force of local water consumption increases due to the social and economic development and the change of residents' consumption structure. During the study period, the pressure on the water system increased due to the increase in driving indicators, while the status of water resources continued to decline[34]. Vannevel, R. embed The DPSIR framework into The Pentatope Model for Environmental Analyses and use the Governance by Actor - Subject Impact Assessment (GASI) as the interface. A tool for water treatment of PTM-GASI-DPSIR is proposed[35]. From the research of these scholars, we affirm the contribution of DPSIR framework in water resources assessment. From these studies, I can learn the logical thinking of analyzing such problems, and find the change characteristics of the research object from the two parts of the subsystem and the whole. In the actual research and application, it is easy to get the advantages and disadvantages of the research object in the subsystem and the whole. This is very helpful for water resources planning and management.
However, with the deepening of the research on this kind of issues, we find that the level of water resource carrying capacity in a region is not only affected by the difference in natural resource endowment and the driving force of water consumption, but also affected by local management measures. The impact of management on water resources system is omni-directional, and management will affect the driving force, pressure, state and response. The DPSIR framework does not yet incorporate management systems into this analytical framework. This does not reflect well on the role of government authorities in water management. If the management system is included in this framework, from the perspective of scientific research, not only increasing management indicators from the evaluation system, but also making the evaluation dimension of water system security and water resources carrying capacity wider, more comprehensive evaluation, and more objective and accurate evaluation results. Moreover, it is of great value to further understand the internal relationship of each system of water resources and propose solutions to the problems of water resources management from the perspective of management. Adding management elements to the DPSIR framework becomes DPSIRM.
Materials and methods
Line no-134: This may be a pressure sub-system. Please make it more clear of the pressure on what?
Response:Thank you for your valuable suggestion. This has been revised and supplemented in the article. The revisions are as follows:
The pressure subsystem (P) refers to the pressure exerted on water resources environment by intentional or unintentional human activities, such as the influence of economic development level on water resources utilization, the change of water consumption of urban residents caused by urbanization development, and the discharge of industrial wastewater.
Results
Line no-231: Please edit the year.
Response:Thank you for your valuable suggestion. The year has been added
The revisions are as follows:
This indicates that the population size, economic development level and urbanization rate of Shiyan City during 2011-2021 are the factors that cannot be ignored to improve the water resources carrying capacity and enhance the water system elasticity of Shiyan City.
Line no-232: Please edit the year writing style.
Response:Thank you for your valuable suggestion. The year writing style has been revised.The revisions are as follows:
The D subsystem was the first obstacle system in 2016 and 2017
Line no- 250: This is not an appropriate way of writing. Authors are requested to edit it like "The finding of first the obstacle ....".
Response:Thank you for your valuable suggestion.Your suggestion is very good, and we have revised it according to your suggestions. The revisions are as follows:
The finding of first the obstacle, between 2011 and 2021 M2 index appeared 8 times, R2 index appeared 2 times, and D2 index appeared 1 time.
Line no-267: This may be one time.
Response:Thank you for your valuable suggestion.Your suggestion is very good, and we have revised it according to your suggestions. The revisions are as follows:
the M2 index appeared one time.
Line no-267: How did the authors calculate the average annual fertilizer application? Please mention in the method section and its source.
Response:Thank you for your valuable suggestion. The average annual fertilizer application per unit of cultivated land is obtained by dividing the total fertilizer application amount by the total cultivated land area. The revisions are as follows:
Table 1. Evaluation system of water resources carrying capacity in Shiyan City
Criterion layer |
Indicator layer |
Properties |
Calculation methods |
Reference |
Driving force (D) |
XD1 Per capita GDP(yuan) |
Positive |
From statistical data |
[38][40] |
|
XD2 density of population |
Negative |
From statistical data |
[38][40] |
|
XD3 urbanization rate |
Negative |
From statistical data |
[38][40] |
Pressure (P) |
XP1 Wastewater discharge per unit of industrial output value (t/ 10,000 Yuan) |
Negative |
Amount industrial wastewater discharge/ Industrial output value |
[39][40] |
|
XP2 Household water consumption (10,000 m3) |
Negative |
From statistical data |
[39] |
|
XP3 Average annual fertilizer application per unit cultivated land (kg/hm2) |
Negative |
Amount fertilizer application/ cultivated area |
[39][40] |
Status (S) |
XS1 Water resources per capita (m³) |
Positive |
Amount of regional water resource/ regional population |
[38][40] |
|
XS2 Water resources per unit area (m³/hm2) |
Positive |
Amount of regional water resource/ regional land area |
[38][40] |
|
XS3 Annual precipitation (100 million cubic meters) |
Positive |
From statistical data |
[41] |
Impact (I) |
XI1 Proportion of guaranteed harvest area of drought and flood in cultivated land (%) |
Positive |
Guaranteed harvest area in drought and flood/ cultivated area |
[42] |
|
XI2 Water quality in line with Class â… ~ â…¢ standard proportion |
Positive |
From statistical data |
[39] |
|
XI3 Forest coverage rate (%) |
Positive |
From statistical data |
[42] |
Response (R) |
XR1 Sewage treatment rate (%) |
Positive |
From statistical data |
[39][40] |
|
XR2 Length of drainage pipe (km) |
Positive |
From statistical data |
[38] |
Management (M) |
XM1 Green coverage rate of built-up areas (%) |
Positive |
The annual built-up green cover area/green cover area |
[38] |
|
XM2 Investment in wastewater treatment (10,000 yuan) |
Positive |
From statistical data |
[40] |
Conclusion
Line no-285: Please insert this whole paragraph in the introduction section.
Response:Thank you for your valuable suggestion.We have inserted this paragraph into the introduction as you suggested.
Line no-290: Please write a short concluding result.
Response:Thank you for your valuable suggestion.We have inserted this paragraph into the introduction as you suggested.We have compressed the number of words in the conclusionThe revisions are as follows:
The DPSIRM framework is used to study the water system security from the perspective of water resources carrying capacity. From 2011 to 2021, the carrying capacity of water resources in Shiyan City is gradually improved, and the safety of water system is gradually improved. The resilience of the water system in Shiyan City is increasing. The management system is the main obstacle subsystem, followed by the driving force system, the response system, the pressure system, the state system, and the influence system. Among the specific factors, the top three obstacle factors are sewage treatment investment, the proportion of guaranteed harvest area in drought and flood, and the average annual fertilizer applied per unit of cultivated land. These factors are the main factors restricting Shiyan City’s water resources carrying capacity improvement.
Line no-290: No need to mention the limitations in the conclusion section.
Response:Thank you for your valuable suggestion. After discussion, members of our research group think that there are still many problems in this area that need to be further studied in the future, and we think it is necessary to write down the limitations of the research.
Reviewer 2 Report
The thematic scope of the manuscript is relevant to the aims and scope of "Water"journal. The research goals, methods and results are presented in a satisfactory way. Below are several points for consideration to improve the paper:
1. First three sentences of the Abstract are redundant.
2. Please describe clearly the scientific added value of this work (combining the evaluation of water resource carrying capacity with the evaluation of factors influencing it?)
3. Please develop the critical analysis of the existing body of knowledge on DPSIR framework.
4. Please justify the use of the entropy-based weight determination.
5. Discussion of the limitations of the study should be further developed.
Minor editing of English language required.
Author Response
- First three sentences of the Abstract are redundant.
Response:Thank you for your valuable suggestion.We delete the first three sentences.
- Please describe clearly the scientific added value of this work (combining the evaluation of water resource carrying capacity with the evaluation of factors influencing it?)
Response:Thank you for your valuable suggestion. The revisions are as follows:
From the perspective of research area, Shiyan city is the core water source area of the South-to-North Water Diversion Project. The carrying capacity of water resources in the core water source area is related to the safety of the local water system and the sustainability of the project. A search of literature from web of science shows that it is extremely rare to study the water resources carrying capacity of the entire Han River basin where Shiyan City is located from the perspective of water resources management. In this study, the use of DPSIRM to study this topic can objectively understand the state of the local water resources, and provide help for the local water resources management. From the perspective of research content, the specific index selection of this study draws on many academic research results, and also combines with the actual situation of local indicators, which can provide some reference for the index selection of the same type of research.
- Please develop the critical analysis of the existing body of knowledge on DPSIR framework.
Response:Thank you for your valuable suggestion. In the literature review section, we added a separate review of the research framework, mainly analyzing the DPSIR framework.The revisions are as follows:
At present, the Driving Force-Pressure-State-Impact-Response (DPSIR) framework is the most widely used in environmental assessment. The DPSIR framework was originally developed by the European Environment Agency (EEA) in 1995 to provide decision-makers with information on environmental indicators, in response to the European Environment Agency's (EEA) proposal on how to proceed with the development of an integrated environmental assessment strategy. DPSIR is an extension of the PSR framework, adding driving force (D) and impact (I). The PSR framework was originally developed by the OECD to organize its work on environmental policy and reporting. These five aspects are logically causally related, which is a very good analytical framework for exploring the relationship between environment and socio-economic activities. This analytical framework was initially a qualitative evaluation system, and later scholars used this evaluation system quantitatively.
Later, some scholars applied DPSIR framework to environmental assessment, ecological security, water resources management, sustainable development, air pollution and other fields. The representative studies on water resources security, water resources management and water system risk using DPSIR framework are as follows. Using the DPSIR(Drive, Stress, State, Impact, Response) method, Borja, A. et al studied the ecological quality and risk of local water bodies in estuaries and coastal waters in the Basque country (northern Spain)[32]. Skoulikidis, N. T. examined the state of the environment of 15 major Balkan rivers within the framework of DPSIR, arguing that the wars, political instability, economic crises of the past decades, combined with administrative and structural constraints, poor environmental planning and inspections, and often a lack of environmental awareness, have put great pressure on the rivers[33]. Sun, S. et al. established an evaluation index of sustainability of water resources use based on the drives-pressure-state-impact-response (DPSIR) model. The sustainable change of water resources system in Bayannur City was evaluated comprehensively by using these indexes. The results show that the driving force of local water consumption increases due to the social and economic development and the change of residents' consumption structure. During the study period, the pressure on the water system increased due to the increase in driving indicators, while the status of water resources continued to decline[34]. Vannevel, R. embed The DPSIR framework into The Pentatope Model for Environmental Analyses and use the Governance by Actor - Subject Impact Assessment (GASI) as the interface. A tool for water treatment of PTM-GASI-DPSIR is proposed[35]. From the research of these scholars, we affirm the contribution of DPSIR framework in water resources assessment. From these studies, I can learn the logical thinking of analyzing such problems, and find the change characteristics of the research object from the two parts of the subsystem and the whole. In the actual research and application, it is easy to get the advantages and disadvantages of the research object in the subsystem and the whole. This is very helpful for water resources planning and management.
However, with the deepening of the research on this kind of issues, we find that the level of water resource carrying capacity in a region is not only affected by the difference in natural resource endowment and the driving force of water consumption, but also affected by local management measures. The impact of management on water resources system is omni-directional, and management will affect the driving force, pressure, state and response. The DPSIR framework does not yet incorporate management systems into this analytical framework. This does not reflect well on the role of government authorities in water management. If the management system is included in this framework, from the perspective of scientific research, not only increasing management indicators from the evaluation system, but also making the evaluation dimension of water system security and water resources carrying capacity wider, more comprehensive evaluation, and more objective and accurate evaluation results. Moreover, it is of great value to further understand the internal relationship of each system of water resources and propose solutions to the problems of water resources management from the perspective of management. Adding management elements to the DPSIR framework becomes DPSIRM.
- Please justify the use of the entropy-based weight determination.
Response:Thank you for your valuable suggestion. We supplement the process of entropy weight method in the aspect introduction section. The revisions are as follows:
The entropy method was used to determine the weights in this study. The main steps are in paper.
- Discussion of the limitations of the study should be further developed.
Response:Thank you for your valuable suggestion. We further supplemented the limitations of the study.The revisions are as follows:
Although our study objectively evaluated the trend of changes in water resource carrying capacity in Shiyan City, we also identified factors that constrain the further improvement of the resilience of the water resource system in Shiyan City from the perspectives of subsystems and specific factors. However, it did not fully consider the cross regional spillover effects in water resource management in surrounding cities. Future research should expand the research area and further study the interactions and impacts between cities. In the selection of indicators in management, because there is no corresponding statistical data, and there is no public multi-year data, only two dimensions are selected in the selection of indicators. This is what needs to be further enriched and perfected in the future research.
Comments on the Quality of English Language
Minor editing of English language required.
Response:Thank you for your valuable suggestion. We further optimized the language expression.
Reviewer 3 Report
The paper is interesting reading but I have some issues with the paper.
1.Regarding the methodology section, additional explanation is required on the methods used for data analysis and collection
2. The paper lacks a proper literature review on DPSIR models.
3. The DIPSRM model is used for this study. How do you determine whether the properties are positive of negative
4 The M-dimension is operationalized by two Green coverage rate of built-up and Investment in wastewater treatment. What about managerial aspects such as expenses, manpower and financial resources?
5. The language needs to be improved
The language needs to be improved. Some sentence do not run fluidly
Author Response
The paper is interesting reading but I have some issues with the paper.
1.Regarding the methodology section, additional explanation is required on the methods used for data analysis and collection
Response:Thank you for your valuable suggestion. The revisions are as follows:
Table 1. Evaluation system of water resources carrying capacity in Shiyan City
Criterion layer |
Indicator layer |
Properties |
Calculation methods |
Reference |
Driving force (D) |
XD1 Per capita GDP(yuan) |
Positive |
From statistical data |
[38][40] |
|
XD2 density of population |
Negative |
From statistical data |
[38][40] |
|
XD3 urbanization rate |
Negative |
From statistical data |
[38][40] |
Pressure (P) |
XP1 Wastewater discharge per unit of industrial output value (t/ 10,000 Yuan) |
Negative |
Amount industrial wastewater discharge/ Industrial output value |
[39][40] |
|
XP2 Household water consumption (10,000 m3) |
Negative |
From statistical data |
[39] |
|
XP3 Average annual fertilizer application per unit cultivated land (kg/hm2) |
Negative |
Amount fertilizer application/ cultivated area |
[39][40] |
Status (S) |
XS1 Water resources per capita (m³) |
Positive |
Amount of regional water resource/ regional population |
[38][40] |
|
XS2 Water resources per unit area (m³/hm2) |
Positive |
Amount of regional water resource/ regional land area |
[38][40] |
|
XS3 Annual precipitation (100 million cubic meters) |
Positive |
From statistical data |
[41] |
Impact (I) |
XI1 Proportion of guaranteed harvest area of drought and flood in cultivated land (%) |
Positive |
Guaranteed harvest area in drought and flood/ cultivated area |
[42] |
|
XI2 Water quality in line with Class â… ~ â…¢ standard proportion |
Positive |
From statistical data |
[39] |
|
XI3 Forest coverage rate (%) |
Positive |
From statistical data |
[42] |
Response (R) |
XR1 Sewage treatment rate (%) |
Positive |
From statistical data |
[39][40] |
|
XR2 Length of drainage pipe (km) |
Positive |
From statistical data |
[38] |
Management (M) |
XM1 Green coverage rate of built-up areas (%) |
Positive |
The annual built-up green cover area/green cover area |
[38] |
|
XM2 Investment in wastewater treatment (10,000 yuan) |
Positive |
From statistical data |
[40] |
- The paper lacks a proper literature review on DPSIR models.
Response:Thank you for your valuable suggestion. The revisions are as follows:
At present, the Driving Force-Pressure-State-Impact-Response (DPSIR) framework is the most widely used in environmental assessment. The DPSIR framework was originally developed by the European Environment Agency (EEA) in 1995 to provide decision-makers with information on environmental indicators, in response to the European Environment Agency's (EEA) proposal on how to proceed with the development of an integrated environmental assessment strategy. DPSIR is an extension of the PSR framework, adding driving force (D) and impact (I). The PSR framework was originally developed by the OECD to organize its work on environmental policy and reporting. These five aspects are logically causally related, which is a very good analytical framework for exploring the relationship between environment and socio-economic activities. This analytical framework was initially a qualitative evaluation system, and later scholars used this evaluation system quantitatively.
Later, some scholars applied DPSIR framework to environmental assessment, ecological security, water resources management, sustainable development, air pollution and other fields. The representative studies on water resources security, water resources management and water system risk using DPSIR framework are as follows. Using the DPSIR(Drive, Stress, State, Impact, Response) method, Borja, A. et al studied the ecological quality and risk of local water bodies in estuaries and coastal waters in the Basque country (northern Spain)[32]. Skoulikidis, N. T. examined the state of the environment of 15 major Balkan rivers within the framework of DPSIR, arguing that the wars, political instability, economic crises of the past decades, combined with administrative and structural constraints, poor environmental planning and inspections, and often a lack of environmental awareness, have put great pressure on the rivers[33]. Sun, S. et al. established an evaluation index of sustainability of water resources use based on the drives-pressure-state-impact-response (DPSIR) model. The sustainable change of water resources system in Bayannur City was evaluated comprehensively by using these indexes. The results show that the driving force of local water consumption increases due to the social and economic development and the change of residents' consumption structure. During the study period, the pressure on the water system increased due to the increase in driving indicators, while the status of water resources continued to decline[34]. Vannevel, R. embed The DPSIR framework into The Pentatope Model for Environmental Analyses and use the Governance by Actor - Subject Impact Assessment (GASI) as the interface. A tool for water treatment of PTM-GASI-DPSIR is proposed[35]. From the research of these scholars, we affirm the contribution of DPSIR framework in water resources assessment. From these studies, I can learn the logical thinking of analyzing such problems, and find the change characteristics of the research object from the two parts of the subsystem and the whole. In the actual research and application, it is easy to get the advantages and disadvantages of the research object in the subsystem and the whole. This is very helpful for water resources planning and management.
However, with the deepening of the research on this kind of issues, we find that the level of water resource carrying capacity in a region is not only affected by the difference in natural resource endowment and the driving force of water consumption, but also affected by local management measures. The impact of management on water resources system is omni-directional, and management will affect the driving force, pressure, state and response. The DPSIR framework does not yet incorporate management systems into this analytical framework. This does not reflect well on the role of government authorities in water management. If the management system is included in this framework, from the perspective of scientific research, not only increasing management indicators from the evaluation system, but also making the evaluation dimension of water system security and water resources carrying capacity wider, more comprehensive evaluation, and more objective and accurate evaluation results. Moreover, it is of great value to further understand the internal relationship of each system of water resources and propose solutions to the problems of water resources management from the perspective of management. Adding management elements to the DPSIR framework becomes DPSIRM.
- The DIPSRM model is used for this study. How do you determine whether the properties are positive of negative
Response:Thank you for your valuable suggestion. The revisions are as follows:
3.2. Data set and source
This study involves the water resources data, ecological environment data, economy and industry data, and other data of Shiyan City, which is derived from the ‘Shiyan Statistical Yearbook’ from 2012 to 2022, Shiyan City StatisticalBulletin. The basis of indicator selection, determination of positive and negative signs, acquisition and calculation of indicator data are shown in Table 1.
Table 1. Evaluation system of water resources carrying capacity in Shiyan City
Criterion layer |
Indicator layer |
Properties |
Calculation methods |
Reference |
Driving force (D) |
XD1 Per capita GDP(yuan) |
Positive |
From statistical data |
[38][40] |
|
XD2 density of population |
Negative |
From statistical data |
[38][40] |
|
XD3 urbanization rate |
Negative |
From statistical data |
[38][40] |
Pressure (P) |
XP1 Wastewater discharge per unit of industrial output value (t/ 10,000 Yuan) |
Negative |
Amount industrial wastewater discharge/ Industrial output value |
[39][40] |
|
XP2 Household water consumption (10,000 m3) |
Negative |
From statistical data |
[39] |
|
XP3 Average annual fertilizer application per unit cultivated land (kg/hm2) |
Negative |
Amount fertilizer application/ cultivated area |
[39][40] |
Status (S) |
XS1 Water resources per capita (m³) |
Positive |
Amount of regional water resource/ regional population |
[38][40] |
|
XS2 Water resources per unit area (m³/hm2) |
Positive |
Amount of regional water resource/ regional land area |
[38][40] |
|
XS3 Annual precipitation (100 million cubic meters) |
Positive |
From statistical data |
[41] |
Impact (I) |
XI1 Proportion of guaranteed harvest area of drought and flood in cultivated land (%) |
Positive |
Guaranteed harvest area in drought and flood/ cultivated area |
[42] |
|
XI2 Water quality in line with Class â… ~ â…¢ standard proportion |
Positive |
From statistical data |
[39] |
|
XI3 Forest coverage rate (%) |
Positive |
From statistical data |
[42] |
Response (R) |
XR1 Sewage treatment rate (%) |
Positive |
From statistical data |
[39][40] |
|
XR2 Length of drainage pipe (km) |
Positive |
From statistical data |
[38] |
Management (M) |
XM1 Green coverage rate of built-up areas (%) |
Positive |
The annual built-up green cover area/green cover area |
[38] |
|
XM2 Investment in wastewater treatment (10,000 yuan) |
Positive |
From statistical data |
[40] |
4 The M-dimension is operationalized by two Green coverage rate of built-up and Investment in wastewater treatment. What about managerial aspects such as expenses, manpower and financial resources?
Response:Thank you for your valuable suggestion. The indicators you mentioned belong to the management side. The main difficulty is that no corresponding statistics can be found, even through the calculation there are no associated statistics. Therefore, in this study, in order to closely focus on this subject, we draw lessons from the existing studies to express management aspects from Green coverage rate of built-up and Investment in wastewater treatment.
- The language needs to be improved
Response:Thank you for your valuable suggestion. We further optimized the language expression.
Round 2
Reviewer 1 Report
The authors edited and modified all the suggestions. I would like to thank you all for your hard work and extensive editing. Now, I suggest you for a minor English language edit.
Reviewer 3 Report
I would like to see more critical reflections on the methodology used and how this influenced the results.
Somehow, I feel there is a lot more relevant references on DPSIR to include in the paper.
Please conduct a check on grammar and typos